# Fuzzy Logic as a Strategy for Combining Marker Statistics to Optimize Preselection of High-Density and Sequence Genotype Data

**DOI:** 10.3390/genes13112100

**Published:** 2022-11-11

**Authors:** Ashley Ling, El Hamidi Hay, Samuel E. Aggrey, Romdhane Rekaya

**Affiliations:** 1USDA Agricultural Research Service, Fort Keogh Livestock and Range Research Laboratory, Miles City, MT 59301, USA; 2Department of Poultry Science, The University of Georgia, Athens, GA 30602, USA; 3Institute of Bioinformatics, The University of Georgia, Athens, GA 30602, USA; 4Department of Animal and Dairy Science, The University of Georgia, Athens, GA 30602, USA; 5Department of Statistics, The University of Georgia, Athens, GA 30602, USA

**Keywords:** SNP preselection, genomic prediction, high-density genotypes, sequence genotypes, fuzzy logic, fuzzy inference

## Abstract

The high dimensionality of genotype data available for genomic evaluations has presented a motivation for developing strategies to identify subsets of markers capable of increasing the accuracy of predictions compared to the current commercial single nucleotide polymorphism (SNP) chips. In this simulation study, an algorithm for combining statistics used in the preselection and prioritization of SNP markers from a high-density panel (1.3 million SNPs) into a composite “fuzzy” ranking score based on a Sugeno-type fuzzy inference system (FIS) was developed and evaluated for performance in preselection for genomic predictions. F${}_{\mathrm{ST}}$ scores, and *p*-values were evaluated as inputs for the FIS. The accuracy of genomic predictions for fuzzy-score-preselected panel sizes of 1–50 k SNPs ranged from −0.4–11.7 and −0.3–3.8% higher than F${}_{\mathrm{ST}}$ and *p*-value preselection, respectively. Though gains in prediction accuracies using only two inputs to the FIS were modest, preselection based on fuzzy scores yielded more accurate predictions than both F${}_{\mathrm{ST}}$ scores and *p*-values for the majority of evaluated panel sizes under all genetic architectures. FIS have the potential to aggregate information from multiple criteria that reflect SNP-trait associations and biological relevance in a flexible and efficient way to yield higher quality genomic predictions.

## 1. Introduction

The availability of high-density (HD) and whole genome sequence (WGS) data has increased in recent years to allow their practical use in genomic evaluations. It was initially expected that prediction accuracies would increase using such data as a result of better tracking of the sources of genetic variation [1]. However, practical results demonstrate little to no advantage relative to low- and moderate-density single nucleotide polymorphism (SNP) panels currently used in genomic evaluations [2]. This failure is likely due to a combination of excessive multicollinearity between SNP/variant predictors [3] and overparameterization of the association model [4].

SNP marker preselection has the potential to improve the quality of genomic predictions by reducing multicollinearity and overparameterization and by increasing the biological relevance of markers included in the association model or in the calculation of the genomic relationship matrix. Several published studies have evaluated single preselection criteria, including *p*-values [5], estimated marker effects [6], F${}_{\mathrm{ST}}$ scores [7,8], and annotation missense status [9]. In each of these studies, prediction accuracies were better than genome-wide SNP panels only when one or a few variants had a large contribution to the genetic variance of the trait for both simulated and real data.

The limited success of single-criteria preselection to improve prediction accuracies is likely to be explained by the limited sensitivity of any single criteria to distinguish between relevant and non-relevant markers. All criteria will be subject to false positives, and the impact of such markers on predictions is unlikely to be null. Ling et al. (2021) [10] demonstrated that while the SNP preselection by the estimated effect results in less false positives and more genetic variance explained than F${}_{\mathrm{ST}}$ preselection, the false positives selected based on estimated effect have a greater negative impact on predictions and can ultimately offset the benefit from capturing more genetic variance. Based on these observations, it is reasonable to hypothesize that a composite score combining these two preselection criteria may harness the benefits of both to ultimately yield more accurate genomic predictions. Ref. [11] proposed a composite score for ranking variants from WGS data based on association and biological criteria and found that when preselected variants were combined with a genome-wide 630 k SNP panel, differences in genomic prediction accuracies between high- and low-ranking variants were statistically significant for several economically important milk traits in Holsteins.

It has been demonstrated that the increase in the accuracy of genomic predictions in the presence of HD marker data requires a balance between the fraction of the genetic variance explained by the prioritized SNPs and the number of preselected markers [8]. Increasing the latter will often increase the portion of genetic variance explained but not the accuracy of the genomic predictions. Thus, if WGS variants fail to significantly improve the prediction accuracies solely by explaining greater genetic variance, then increasing accuracies must rely instead on decreasing the bias in genomic predictions.

Fuzzy logic was initially proposed by Zadeh (1965) [12] as a strategy to mimic the human approach in understanding data-generating processes and logical decision making, where factors in the decision-making process may be subject to vagueness and uncertainty. It has subsequently been applied to a variety of practical applications, from inventory control [13] to analysis of landslide susceptibility [14], due to its flexibility and efficiency in modeling nonlinear processes [15]. In human medical research, fuzzy logic algorithms have been evaluated for their use in disease classification based on genomic markers [16,17]. Fuzzy logic models have also found various applications in agriculture. They have been extensively utilized in dairy cattle production systems, as automated systems have become increasingly utilized [18]. Shahinfar et al. (2012) [19] evaluated the feasibility of estimating breeding values in dairy cattle based on parental estimated breeding values and own performance. More recently, fuzzy logic models have been used to predict phenotypic performance in crop hybrids based on parental line performance data [20] and genomic markers [21]. Fuzzy logic models are a computationally efficient and flexible tool for decision making based on multiple criteria.

A fuzzy inference system (FIS) may be particularly suitable for genotype data and genomic prediction analyses, as it is generally assumed that the vast majority of markers available are in imperfect linkage disequilibrium (LD) with causative variants rather than they themselves having a causative effect on the trait(s) of interest. While thresholds for assessing the amount of useful LD between a SNP and causative variant have been proposed [22], these are clearly somewhat arbitrary and the usefulness of information carried by a particular SNP marker with regards to causative variants is more likely to fall in a gray area between true information and noise. However, it is clear that if the loci of the causative variants were known then the problem that originally motivated the use of SNPs would be solved and there would be no purpose for SNP preselection. Instead, criteria that reflects association of SNPs with a particular trait, and by extension the potential LD with a causative variant, could be used in a fuzzy model. In this simulation study, we evaluate the potential of a Sugeno-type fuzzy logic model [23] to combine *p*-values and F${}_{\mathrm{ST}}$ scores into a composite score for SNP preselection, which to our knowledge has not previously been evaluated as a SNP preselection tool for the prediction of the genomic breeding values of complex traits.

## 2. Materials and Methods

Fuzzy logic is often used in assessing uncertainty associated with a decision-making process. It provides computers with the ability to compare alternatives in a similar manner to the reasoning process of humans. For example, when comparing the daily milk production of two cows producing 40 and 41 kg, a crisp classification system based solely on phenotypic selection for high milk yield will conclude that the daily productions are different and the cow producing 41 kg is a better selection candidate. Realistically, a reasonable farmer would be more likely to use a “soft” approach by balancing the milk production with additional important factors that are not fully assessed (e.g., feed efficiency and health factors) and the final decision would potentially be different from the crisp classification. A FIS is a process that formalizes this strategy and consists of three main components: (1) fuzzification of the inputs, (2) rule evaluation, and (3) defuzzification to achieve a final output. Conceptually, the final fuzzy outputs in this example would not be dissimilar to a selection index where performance across multiple traits is accounted for, though the fuzzy algorithm notably differs from how selection indices are constructed.

Fuzzification of the inputs involves constructing for each input class a fuzzy set, A=(U,m); where *U* is a set that falls within the sample space of a given input, *x*; and *m* is a membership function that maps *x* to membership in *U* with m(x)∈[0,1]. Figure 1 illustrates the mapping of the inputs used in this study (*p*-values associated with estimated SNP effects and marker F${}_{\mathrm{ST}}$ scores) using piecewise linear functions as a membership function (*m*). It is worth noting that a given input value may have partial membership in adjacent classes, hence the fuzziness of the set, under the constraint that the sum of all memberships are less than or equal to one.

The rule evaluation step maps the underlying association between the input and output of the underlying system. Each set of inputs must be evaluated according to a rule base. The rule base follows a generalized structure of,
$$\mathrm{if}(\mathrm{Input}\;1\;\mathrm{is}\;x_{i1}\wedge \mathrm{Input}\;2\;\mathrm{is}\;x_{i2}\wedge \cdots \wedge \mathrm{Input}\;t\;\mathrm{is}\;x_{it})\;\mathrm{then}\;z_{i}=a_{i}^{\prime }x_{i},$$
where $z_{i}$ is the true output for individual *i*, modeled as a linear combination of the vector of input parameters ($x_{i}$); $a_{i}$ is a vector of unknown coefficients; and ∧ stands for the Boolean operator AND. Following the basic fuzzy logic approach formulation, the rule base is constructed according to principles of Boolean logical operators. The choice of a specific Boolean operator has an additional impact on how the truth values are aggregated in the defuzzification step. Each input that factors into a particular rule is associated with a membership in the membership function (*m*) for that rule, yet only one of these membership values will be associated with the resulting truth output. Which of the membership values is associated with the truth value depends on the choice of the Boolean operator. The minimum (maximum) membership value will be selected using the Boolean operator AND (OR). Table 1 shows the rules and corresponding parameters used in this study, where $b_{0}$ is the intercept and $b_{1}$ and $b_{2}$ are the coefficients corresponding to F${}_{\mathrm{ST}}$ scores and *p*-values, respectively.

The defuzzification step is the process used to identify the best crisp outcome given the fuzzy set. It consists of generating a single score (*y*) through the aggregation of the output of the different fuzzy sets,
$$\frac{\sum_{i=1}^{p}m_{i}z_{i}}{\sum_{i=1}^{p}m_{i}},$$
where *p* is the number of rules in the rule base. It can be observed here how the construction of the rules and choice of Boolean operator ultimately influence the final output. Since the Boolean operators AND and OR result in *y* being weighted by the smallest and largest membership values, respectively, for a given rule, the contribution of a particular rule to the final output will vary considerably depending on the choice of operator. It would be reasonable to suggest that alternative strategies to collapsing the input memberships into a membership for the truth value could be used, for example an average, but this was not investigated in the present study.

Figure 2 shows the output space of the crisp fuzzy scores for the range of inputs observed in this study. The fuzzy algorithm was able to accommodate a highly nonlinear output space for the fuzzy scores despite the membership functions and rule base having linear components.

F${}_{\mathrm{ST}}$ scores and *p*-values were chosen as the preselection criteria and fuzzy inputs in this study due to the straightforwardness of calculating these statistics in a simulation environment, but it should be noted that in real data there are many additional criteria that have sound justification for being indicative of biological relevance and could potentially be integrated into a fuzzy model for preselection.

*p*-values were calculated jointly using a single-step genome-wide association study approach with the BLUPF90 suite of programs, as described in [24], where genomic predictions are first calculated using a single-step GBLUP (ssGBLUP) model and SNP effects are then jointly calculated from the genomic predictions,
$$\hat{a}=\frac{1}{2\sum_{i=1}^{m}p_{i}q_{i}}Z^{\prime }G^{-1}\hat{u},$$
where $Z^{\prime }$ is the transpose of the *n* × *m* matrix of genotypes, $G^{-1}$ is the inverse of the genomic relationship matrix, $\hat{u}$ is the vector of solutions of genotyped individuals from a ssGBLUP model, and $p_{i}$ and $q_{i}$.

The SNP variances are calculated according to,
$$Var\left({\hat{a}}_{i}\right)=\frac{1}{\sum_{i=1}^{m}2p_{i}q_{i}}z_{i}^{\prime }G^{-1}\left(G\sigma_{u}^{2}-C^{u_{2}u_{2}}\right)G^{-1}z_{i}\frac{1}{\sum_{i=1}^{m}2p_{i}q_{i}}.$$

It should be noted that while the equivalency of GBLUP and SNPBLUP has been demonstrated under certain assumptions [25], SNPBLUP and ssGBLUP are not strictly equivalent due to the inclusion of information from non-genotyped animals in the latter.

F${}_{\mathrm{ST}}$ scores were calculated as in [7]. With this strategy, subpopulations are formed from the extreme tails of the distribution of phenotypes. Groups of individuals with highly divergent phenotypes are expected on average to carry different alleles at loci that contribute variation to the phenotype of interest and F${}_{\mathrm{ST}}$ is a natural measure of such differences,
$$F_{ST}=\frac{H_{T}-H_{S}}{H_{T}},$$
where $H_{T}=2p_{T}q_{T}$, $H_{S}=\frac{H_{S1}n_{S1}+H_{S2}n_{S2}}{n_{S1}+n_{S2}}$ and $H_{Si}=2p_{Si}q_{Si}$, $p_{T}$ and $q_{T}$ are major and minor allele frequencies, respectively, in the combined populations; $p_{Si}$ and $q_{Si}$ are major and minor allele frequencies, respectively, in the $i_{th}$ subpopulation; and $n_{Si}$ is the number of individuals in the $i_{th}$ subpopulation.

While separate breeding populations are traditionally used for calculations of F${}_{\mathrm{ST}}$ and it would seem natural to choose distinct breeding populations with divergent phenotypes (e.g., breeds with historically different selection goals), such an approach has the drawback of capturing deviations in allele frequencies at loci affecting all traits for which the populations differ on average. By grouping individuals from a single population by extreme phenotype, deviations in allele frequencies at unrelated variants will be driven only by correlated traits and random sampling.

Simulated datasets were generated using QMSim [26] for three scenarios that differed in the number of the causative loci contributing variation to the trait of interest (300, 1 k, or 5 k). Table 2 shows parameters specified for the simulation. Each simulation consisted of ten replicates and three sequential populations. First, a historical population (HP) under random mating was generated to establish mutation-drift balance and LD. None of these individuals had data or pedigree recorded. For the second population, 500 males and 4000 females were randomly selected from HP to be founders. This population underwent phenotypic selection and the number of dams retained expanded by 5% each generation. All individuals had data and pedigree recorded without error. From this population, 100 males and 6000 females were selected to be founders of the most recent population. It consisted of 5 non-overlapping generations under selection and 3500 individuals per generation. Selection was based on pedigree EBVs. All individuals had data and pedigree and a random 5000 individuals from generations 1 to 4 and 1000 individuals from generation 5 had genotypes.

Preselection was based on random selection, *p*-values, F${}_{\mathrm{ST}}$ scores, or fuzzy scores as criteria. The predictive power of preselected SNP panels of size 1, 2, 3, 4, 5, 10, 20, 30, 40, and 50 k was evaluated using a ssGBLUP model, as implemented in BLUPF90 [24].

The simulated genome consisted of 1.3 million SNPs randomly distributed across 29 chromosomes. There were either 300, 1 k, or 5 k causative variants randomly distributed across either only 14 or all 29 of the chromosomes; in the prior scenario, SNPs on 15 chromosomes did not segregate with any causative loci and was designed in such a way to allow investigation of how noninformative SNPs affect genomic predictions and sensitivity of different preselection criteria.

Genomic prediction accuracies were calculated as the correlation between true (u) and estimated ($\hat{u}$) breeding values. Bias of SNP associations was measured as the deviation of the regression coefficient of u on $\hat{u}$ ($b_{1,u|\hat{u}}$) from one, with values less and greater than one indicating over- and underdispersion, respectively. If the estimated effect of a SNP (or its association with the trait) is not consistent from the training to the validation population, then its contribution to the validation genomic predictions will create a variation that is not present in the TBVs and results in overdispersion of the estimates ($b_{1,u|\hat{u}}<1$).

The amount of genetic variance explained (GVE) by preselected SNPs was quantified as the variance of validation predictions, var($\hat{u}$). The genomic prediction accuracy can be expressed as a function of the regression of true on estimated breeding values, $b_{1,u|\hat{u}}$, and the ratio of the standard deviations of u and $\hat{u}$,
(1)$$cor\left(u,\hat{u}\right)=b_{1,u|\hat{u}}\frac{sd(\hat{u})}{sd(u)}.$$

## 3. Results

In the simulated genome, only 14 of the 29 chromosomes harbored causative variants, leaving the SNPs on 15 of the chromosomes definitively unlinked with any causative loci. Figure 3 shows the proportion of SNPs that are linked (blue) or not linked (red) selected by each of the marker prioritization criteria. As expected, random preselection resulted in the most unlinked markers for all genetic architectures. F${}_{\mathrm{ST}}$ scores preselected more unlinked markers than *p*-values for all genetic architectures and SNP panel sizes. Fuzzy preselection resulted in the least number of unlinked markers for all preselected panels and genetic architectures, though the differences between *p*-value and fuzzy preselection are relatively modest. Interestingly, the advantage of the fuzzy scores relative to *p*-values in discriminating between linked and unlinked markers seemed to increase as the genetic architecture increases in complexity, with the percent difference in the number of linked markers selected by fuzzy scores ranging from 0.17 to 2.77, 0.67 to 2.57, and 2.49 to 5.40% higher than *p*-values for 300, 1 k, and 5 k causative variants, respectively.

It may be useful to further compare how markers that are linked and unlinked with causative variants are commonly (or not) selected across preselection criteria. Table 3, Table 4 and Table 5 present the number of SNP markers that are unique to a particular criterion and the proportion of those markers that are linked with causative variants. Table 6, Table 7 and Table 8 show the same information but for SNP that are shared between two or more criteria. Between 15 to 25, 14 to 32, and 16 to 31% of the SNPs preselected based on fuzzy scores for 300, 1 k, 5 k causative variants, respectively, were not present in the equivalent-sized panels for F${}_{\mathrm{ST}}$ scores and *p*-values and a higher proportion of these SNPs were linked with causative variants for all genetic architectures. As expected, a high proportion of the SNPs that were selected by all criteria for a given panel size were linked with causative variants; while the proportion linked decreased as the panel size increased due to an expected loss of sensitivity, the proportion linked was still considerably higher than among SNPs selected by only one or two of the criteria, with the notable exception of markers that were selected by both F${}_{\mathrm{ST}}$ scores and *p*-values but not fuzzy scores. Greater than 95% of SNPs were linked with causative variants for all genetic architectures for the latter case, indicating that there may be a more optimal set of fuzzy parameters that would include these SNPs in the fuzzy-preselected panels.

While it is not straightforward how to quantify the relevance of a particular SNP in a scenario where all markers are potentially linked and in LD with a causative variant(s), if the overlap in SNP selected by each criteria is similar to the scenarios where a portion of the SNPs are definitively not linked with any causative variants, then it may be reasonable to expect that the fuzzy scores are robust across genetic architectures for preselection. Table 9 shows the number of SNPs unique to one criteria or shared across two or all criteria for a genetic architecture of 1 k causative variants that are distributed across all 29 chromosomes. While this scenario is expected to be more realistic with regards to the genetic architecture of complex traits in real populations, it is not straightforward how to define the relevance of the preselected SNPs. However, it can be observed that the distribution of SNPs across each criteria in Table 9 is quite similar to that observed under the scenarios, where a portion of the markers are definitively not linked with causative variants (and therefore have minimal relevancy for trait-specific predictions), so it may be reasonable to expect that the relevancy of SNPs selected under the more realistic genetic architectures may also be similar to what was measured under the less realistic genetic architectures.

Figure 4A presents the trends in accuracy for each of the preselection criteria across the different SNP panel sizes. For each of the genetic architectures, fuzzy-score-based prioritization yielded higher accuracies compared to the other preselection criteria. In fact, the accuracy using the fuzzy approach was higher by up to 15.3, 7.9, and 2.4 under the 5 k causative variant scenario; 29.9, 8.9, and 3.5 under the 1 k causative variant scenario; and 44.5, 10.6, and 4.57% under the 300 causative variant scenario relative to random, F${}_{\mathrm{ST}}$, and *p*-value preselection methods, respectively. The largest advantage for the fuzzy score preselection relative to random and F${}_{\mathrm{ST}}$ preselection was when the size of the prioritized marker set (panel size) was 1 k and this difference tended to decrease as the panel size grew with the exception of the scenarios of 5 k causative variants for random preselection (fuzzy preselection accuracy was 0.5% lower) and 300 causative variants for F${}_{\mathrm{ST}}$ score-based prioritization (fuzzy approach accuracy was 0.7% lower). Accuracies for *p*-value preselection did not exceed those of fuzzy preselection for any of the genetic architectures or evaluated panel sizes.

Figure 4B shows $b_{1,u|\hat{u}}$ on the horizontal axis, var($\hat{u}$) on the vertical axis, and a gradient showing values of accuracy that correspond to the range of values. Additionally, the values of $b_{1,u|\hat{u}}$ and var($\hat{u}$) that are observed for each preselection criteria and panel size are plotted to show how each preselection criteria moves through the accuracy parameter space as the SNP panel size increases. For all genetic architectures, random preselection increased the variance of the validation predictions slowly but the overdispersion decreased rapidly as the panel size increased. The opposite trend was observed for *p*-value preselection; it tended to initially capture more variance in the predictions and increase this variance at a faster rate, but predictions based on *p*-value preselection are the most over-dispersed. In the case of 300 causative variants, this overdispersion even increased, explaining why under this genetic architecture the *p*-value accuracy trends downward for panel sizes larger than 3k SNPs. F${}_{\mathrm{ST}}$ scores tended to initially capture similar genetic variation as random preselection but be less over-dispersed; while this dispersion did not decrease as quickly as observed with random preselection with increasing SNP panel size, the variation of the predictions tended to increase more quickly, resulting in higher accuracies for F${}_{\mathrm{ST}}$ scores under 300 and 1 k but not 5 k causative variants.

Similar trends in accuracies, $b_{1,u|\hat{u}}$, and var($\hat{u}$) for fuzzy scores, F${}_{\mathrm{ST}}$ scores, and *p*-values were observed under the more realistic scenario where causative variants were distributed across all chromosomes, as shown in Figure 5. Fuzzy preselection outperformed random, F${}_{\mathrm{ST}}$, and *p*-value preselection by up to 39.7, 7.8 and 3.8, respectively, for 300 causative variants; 24.6, 11.7 and 1.9 for 1 k causative variants; and 6.15, 7.7 and 1.9% for 5 k causative variants. When there were 5 k causative variants, random preselection yielded notably better predictions than the other preselection criteria, outperforming fuzzy preselection for all panels larger than 1 k SNPs by up to 5.3%.

## 4. Discussion

Much of the optimism around the use of HD marker panels and WGS data was motivated by the expectations that such data would vastly increase the percentage of the genetic variation captured and ultimately the accuracy of genomic selection. However, both simulation and real-data-based results have demonstrated little benefit using such data in genomic predictions. Equation (Equation 1) shows that genomic prediction accuracies are influenced by two factors: (1) the proportion of the true genetic variance captured in the validation predictions, and (2) the dispersion of validation predictions relative to the true additive genomic values. Increasing the proportion of true genetic variance captured will not necessarily improve genomic prediction accuracies if the validation predictions become more over-dispersed $b_{1,u|\hat{u}}$.

This is evidenced by the trends of accuracy, GVE, and overdispersion under the 5 k QTL genetic architecture. In such a case, *p*-value-based preselection yielded lower genomic prediction accuracies than all other criteria for SNP panels larger than 20 k despite explaining the largest percentage of the genetic variation. In fact, random preselection explained the least genetic variation of all criteria yet yielded the highest accuracies for SNP panels larger than 10 k. These results demonstrate that maximizing the genetic variation captured by a set of SNPs is not necessarily ideal if there is an accumulation of bias in the SNP associations, as *p*-value preselection has a tendency to do that, given that it consistently yielded the greatest over-dispersion of all criteria evaluated.

The performance of random preselection relative to the non-random preselection criteria improved as the number of causative variants increased. It would be reasonable to expect that random preselection might capture more genetic variance by more evenly covering the genome, where non-random criteria might instead saturate regions with causative variants with the strongest signals and consequently fail to capture genetic variance from causative variants with a smaller contribution; however, Figure 2 shows that random preselection in fact captures the least amount of genetic variation across all genetic architectures.

As the complexity of the genetic architecture increases, the maximum contribution to the genetic variance of a single variant is reduced and the sensitivity of non-random preselection criteria to distinguish true from spurious signals becomes more limited.

*p*-values and F${}_{\mathrm{ST}}$ scores followed opposing trends in accuracy, genetic variance explained, and SNP preselection bias. *p*-values tended to capture a larger portion of the genetic variance but result in more over-dispersed genomic predictions, while F${}_{\mathrm{ST}}$ captured consistently less genetic variation but yielded genomic predictions that were less over-dispersed.

A primary objective of designing a composite preselection score using FIS was to increase accuracies relative to those of the individual preselection criteria used as inputs to the FIS. While fuzzy preselection did not universally yield higher accuracies than *p*-value and F${}_{\mathrm{ST}}$ score preselection, there were certain points at which it performed better and this demonstrates that it is feasible to combine multiple criteria in an index to capture the strengths of each and improve predictions. Accomplishing this depends on preserving or increasing genetic variation captured while minimizing the accumulation of bias in SNP associations.

Though ultimately none of the preselection criteria yielded better predictions than the use of all 1.3 million SNPs, very similar accuracies were obtained using preselected panels of 50 k SNPs and smaller, which may be useful in analyses when it is not feasible to routinely use SNP panels of such large dimension.

Although a HD panel was used in this simulation study (1.3 million markers), the reality of the genomic field in livestock and plant applications is that several million markers/variants are available from whole genome sequencing at different depths. This dramatic increase in the number of variants will undoubtably increase the accumulated bias in variant associations using single criteria for prioritization. The results of this study seem to indicate that an aggregated preselection criterion, such as the FIS approach, may be adequate for such data, especially if further prioritization criteria such as annotation, functional, and expression information associated with the considered SNPs/variants is available.

## 5. Conclusions

A large amount of WGS data is being generated in several livestock applications largely for potential use in genomic evaluations. Currently, such information has not led to a meaningful increase in accuracy compared to low/moderate commercial marker panels. Several marker prioritization methods were proposed, but their performance were inconsistent across applications with slight to no improve in accuracy. In spite of our understanding of the relationship between the portion of the genetic variance explained by the preselected markers and the bias induced by the prioritization of false positives (markers non-linked to causal variants), we were unable to find the optimum balance between these two opposing forces to maximize accuracy. In this study, a fuzzy logic approach was presented to concatenate two prioritization criteria into a unique score to prioritize relevant markers with the expectation of reducing the number false positives tagged by each individual criteria separately. Under varying marker panel size and genetic complexity (number of causal variants), the FIS approach seems to perform generally better than the single criteria approaches. We believe that the performance of the FIS approach could be improved with the inclusion of already available additional prioritization information such as annotation, functional, and expression data. While the results presented here are promising, only simulated data was evaluated, the generation of which necessitates making simplified assumptions regarding the biology of the genome and its role in influencing phenotypes. Future research regarding this methodology will include its application to real data to evaluate the robustness of its performance in SNP selection for genomic predictions as well as the investigation of algorithms for estimation and tuning of FIS model parameters. The FIS approach presents a flexible method to combine noisy/uncertain information to derive reasonable decision-making outputs.

## Figures and Tables

**Figure 1 genes-13-02100-f001:**
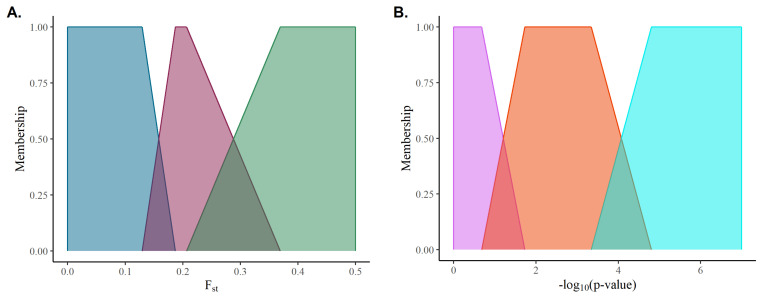
Membership functions for (**A**) F${}_{\mathrm{ST}}$ scores and (**B**) −log_10_ (*p*-value).

**Figure 2 genes-13-02100-f002:**
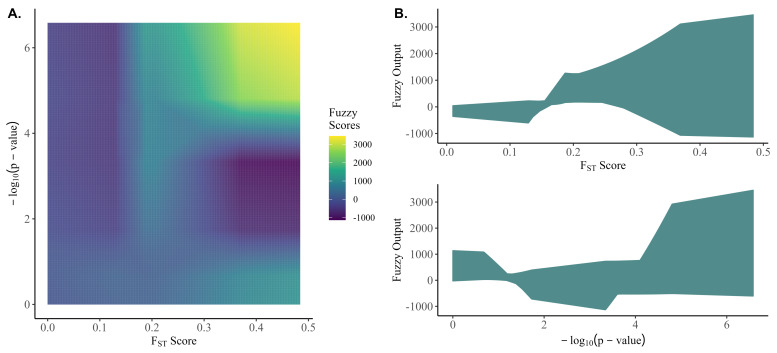
The fuzzy output space for the observed range of F${}_{\mathrm{ST}}$ scores and −log_10_ (*p*-value).

**Figure 3 genes-13-02100-f003:**
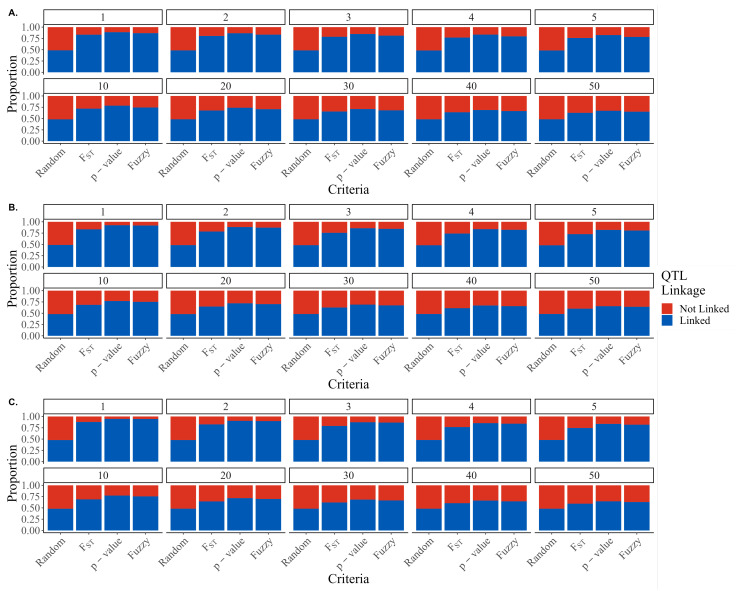
Proportion of SNPs linked with causative variants selected by random, F${}_{\mathrm{ST}}$ scores, *p*-values, or fuzzy scores for (**A**) 5 k, (**B**) 1 k, and (**C**) 300 QTL.

**Figure 4 genes-13-02100-f004:**
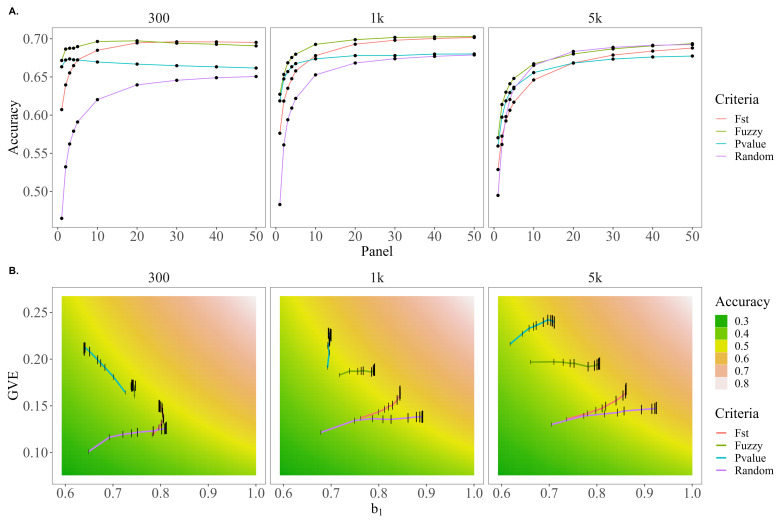
Trends in accuracy, dispersion, and variance of EBVs for random, F${}_{\mathrm{ST}}$ score, *p*-value, or fuzzy score preselection when only 14 of 29 chromosomes harbor causative variants under genetic architectures of 100, 1 k, and 5 k QTL. (**A**) The trends in accuracy for random, F${}_{\mathrm{ST}}$, *p*-value, or fuzzy score preselected panel sizes of 1–50 k SNPs. (**B**) A comparison of how the bias of EBVs ($b_{1}$, *x*-axis) and genetic variance captured by EBVs (var($\hat{u}$), *y*-axis) for random, F${}_{\mathrm{ST}}$, *p*-value, or fuzzy score preselected panel sizes of 1–50 k SNPs (panel size reflected by height of black bar) move through the parameter space of the genomic prediction accuracy (cor(u,$\hat{u}$), gradient background).

**Figure 5 genes-13-02100-f005:**
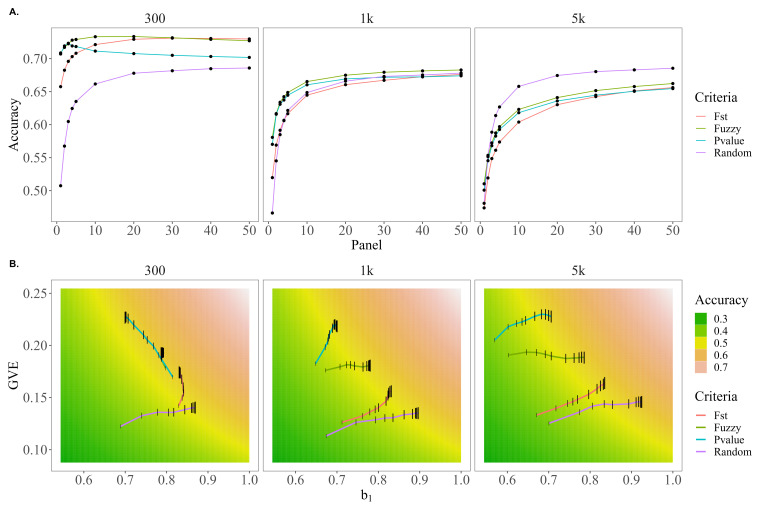
Trends in accuracy, dispersion, and variance of EBVs for random, F${}_{\mathrm{ST}}$ score, *p*-value, or fuzzy score preselection when all chromosomes harbor causative variants. (**A**) The trends in accuracy for random, F${}_{\mathrm{ST}}$, *p*-value, or fuzzy score preselected panel sizes of 1–50 k SNPs. (**B**) A comparison of how the bias of EBVs ($b_{1}$, *x*-axis) and genetic variance captured by EBVs (var($\hat{u}$), *y*-axis) for random, F${}_{\mathrm{ST}}$, *p*-value, or fuzzy score preselected panel sizes of 1–50 k SNPs (panel size reflected by height of black bar) move through the parameter space of the genomic prediction accuracy (cor(u,$\hat{u}$), gradient background).

**Table 1 genes-13-02100-t001:** Fuzzy rules and corresponding rule parameters.

Input Category		Rule Parameters		
**F** ${}_{\mathrm{ST}}$	$-\log_{10}\left(p-\mathrm{Value}\right)$	$b_{0}$	$b_{1}$	$b_{2}$
1	1	−25.68	15.32	8.20
1	2	15.36	−16.79	−9.49
1	3	17.66	−20.66	−5.43
2	1	−0.14	11.18	−6.28
2	2	−20.81	2.73	20.80
2	3	−21.33	−18.67	24.89
3	1	12.81	23.25	−7.86
3	2	−1.82	−5.71	−25.57
3	3	57.42	59.51	57.58

**Table 2 genes-13-02100-t002:** Simulation parameters for QMSim.

*Genomic Parameters*	
N QTL	300 or 1000 or 5000
N SNP	1,300,000
N Chromosomes	29
Have QTL	14 or 29
No QTL	15 or 0
Distribution of QTL Allele Effects	Gamma, Shape = 0.4
Trait Heritability ($h^{2}$)	0.3
*Historical Population (HP) Parameters*	
Number of Generations	4010
Number of Individuals in First 4000 Generations (Males/Females)	400/400
Number of Individuals in Last Generation (Males/Females)	500/4000
*Population to Establish Linkage Disequilibrium (LDP) Parameters ${}^{a}$*	
Number of Generations	20
Number of Founder Individuals (Males/Females)	500/4000
Sire Replacement Rate	0.5
Dam Replacement Rate	0.2
Selection Design	Phenotype
Culling Design	Age
Mating Design	Random
Number of Progeny per Mating	1
*Population Under Evaluation (EP) ${}^{b}$*	
Number of Generations	5
Number of Founder Individuals (Males/Females)	100/6000
Sire Replacement Rate	0.5
Dam Replacement Rate	0.2
Selection Design	EBVs
Culling Design	Age
Mating Design	Random
Number of Progeny per Mating	1

^a^ Founders from last generationof HP; ^b^ Founders from second-to-last and last generations of LDP.

**Table 3 genes-13-02100-t003:** The number of unique SNPs and proportion of unique SNPs that are linked with causative variants for F${}_{\mathrm{ST}}$ score, *p*-value, and fuzzy score preselection under a genetic architecture of 5 k causative variants.

	Unique to One Criteria					
	F${}_{\mathrm{ST}}$ Scores		*p*-Values		Fuzzy Scores	
Panel Size (k)	N	Proportion Linked	N	Proportion Linked	N	Proportion Linked
1	640	0.780	383	0.823	163	0.857
2	1323	0.753	899	0.773	527	0.824
3	2006	0.732	1466	0.748	921	0.808
4	2621	0.712	2046	0.734	1266	0.792
5	3121	0.699	2621	0.720	1486	0.781
10	4849	0.636	5560	0.681	1958	0.738
20	8980	0.588	10,936	0.646	3400	0.689
30	13,113	0.564	15,893	0.622	4727	0.650
40	17,150	0.551	20,581	0.604	5928	0.626
50	21,075	0.540	25,062	0.591	7033	0.604

**Table 4 genes-13-02100-t004:** The number of unique SNPs and proportion of unique SNPs that are linked with causative variants for F${}_{\mathrm{ST}}$ score, *p*-value, and fuzzy score preselection under a genetic architecture of 1 k causative variants.

	Unique to One Criteria					
	F${}_{\mathrm{ST}}$ Scores		*p*-Values		Fuzzy Scores	
Panel Size (k)	N	Proportion Linked	N	Proportion Linked	N	Proportion Linked
1	641	0.759	226	0.844	190	0.899
2	1345	0.713	631	0.769	543	0.932
3	2043	0.686	1182	0.752	964	0.803
4	2630	0.666	1738	0.736	1271	0.777
5	3111	0.647	2288	0.723	1463	0.762
10	4800	0.584	5173	0.677	1896	0.719
20	9114	0.550	10,373	0.631	3434	0.665
30	13,390	0.533	15,210	0.605	4838	0.628
40	17,408	0.520	19,815	0.589	5987	0.611
50	21,284	0.512	24,240	0.577	7019	0.595

**Table 5 genes-13-02100-t005:** The number of unique SNPs and proportion of unique SNPs that are linked with causative variants for F${}_{\mathrm{ST}}$ score, *p*-value, and fuzzy score preselection under a genetic architecture of 300 causative variants.

	Unique to One Criteria					
	F${}_{\mathrm{ST}}$ Scores		*p*-Values		Fuzzy Scores	
Panel Size (k)	N	Proportion Linked	N	Proportion Linked	N	Proportion Linked
1	534	0.791	235	0.884	157	0.893
2	1115	0.729	601	0.786	398	0.805
3	1707	0.695	1108	0.762	710	0.784
4	2264	0.668	1639	0.739	989	0.773
5	2752	0.645	2201	0.718	1218	0.759
10	4813	0.585	5037	0.666	1947	0.705
20	9044	0.539	10,398	0.618	3446	0.643
30	13,445	0.523	15,732	0.591	4971	0.618
40	17,660	0.513	20,132	0.575	6319	0.598
50	21,738	0.507	24,695	0.562	7511	0.581

**Table 6 genes-13-02100-t006:** The number of shared SNPs and proportion of shared SNPs that are linked with causative variants for F${}_{\mathrm{ST}}$ score, *p*-value, and fuzzy score preselection under a genetic architecture of 5 k causative variants.

	Shared by Two Criteria							
Panel Size	F${}_{\mathrm{ST}}$ and *p*-Values		F${}_{\mathrm{ST}}$ and Fuzzy		*p*-Values and Fuzzy		Shared by All Criteria	
(k)	N	PL ${}^{a}$	N	PL ${}^{a}$	N	PL ${}^{a}$	N	PL ${}^{a}$
1	38	0.96	258	0.91	515	0.88	65	0.99
2	90	0.95	462	0.88	886	0.86	126	0.97
3	105	0.95	649	0.86	1189	0.85	241	0.96
4	110	0.94	891	0.85	1465	0.84	378	0.93
5	109	0.92	1244	0.83	1745	0.83	526	0.92
10	81	0.93	3683	0.77	2972	0.80	1388	0.88
20	52	0.95	7587	0.71	5631	0.75	3381	0.84
30	35	0.96	11,201	0.68	8421	0.71	5651	0.81
40	31	0.96	14,684	0.66	11,254	0.69	8135	0.79
50	29	0.96	18,058	0.64	14,071	0.67	10,838	0.77

*^a^* The proportion of shared SNPs linked with QTL.

**Table 7 genes-13-02100-t007:** The number of shared SNPs and proportion of shared SNPs that are linked with causative variants for F${}_{\mathrm{ST}}$ score, *p*-value, and fuzzy score preselection under a genetic architecture of 1 k causative variants.

	Shared by Two Criteria							
Panel Size	F${}_{\mathrm{ST}}$ and *p*-Values		F${}_{\mathrm{ST}}$ and Fuzzy		*p*-Values and Fuzzy		Shared by All Criteria	
(k)	N	PL ${}^{a}$	N	PL ${}^{a}$	N	PL ${}^{a}$	N	PL ${}^{a}$
1	100	0.99	136	0.92	551	0.92	123	0.99
2	169	0.97	257	0.86	971	0.90	230	0.98
3	177	0.97	396	0.83	1257	0.88	384	0.96
4	172	0.96	640	0.80	1532	0.86	558	0.95
5	160	0.96	984	0.77	1806	0.84	746	0.94
10	107	0.97	3385	0.72	3012	0.79	1708	0.89
20	85	0.97	7024	0.67	5764	0.74	3778	0.84
30	78	0.98	10,450	0.64	8630	0.71	6083	0.80
40	72	0.98	13,900	0.63	11,492	0.68	8620	0.77
50	69	0.98	17,290	0.61	14,334	0.67	11,357	0.75

*^a^* The proportion of shared SNPs linked with QTL.

**Table 8 genes-13-02100-t008:** The number of shared SNPs and proportion of shared SNPs that are linked with causative variants for F${}_{\mathrm{ST}}$ score, *p*-value, and fuzzy score preselection under a genetic architecture of 300 causative variants.

	Shared by Two Criteria							
Panel Size	F${}_{\mathrm{ST}}$ and *p*-Values		F${}_{\mathrm{ST}}$ and Fuzzy		*p*-Values and Fuzzy		Shared by All Criteria	
(k)	N	PL ${}^{a}$	N	PL ${}^{a}$	N	PL ${}^{a}$	N	PL ${}^{a}$
1	88	0.99	165	0.95	464	0.94	214	0.99
2	165	0.99	368	0.89	882	0.92	353	0.99
3	189	0.98	587	0.84	1186	0.90	517	0.98
4	199	0.98	848	0.81	1474	0.88	689	0.96
5	202	0.97	1186	0.79	1737	0.86	860	0.95
10	155	0.97	3246	0.71	3022	0.81	1786	0.90
20	117	0.99	7069	0.67	5716	0.74	3770	0.84
30	108	0.99	10,508	0.64	8582	0.70	5939	0.81
40	103	0.99	13,916	0.62	11,444	0.67	8321	0.77
50	100	0.99	17,284	0.60	14,327	0.65	10,878	0.75

*^a^* The proportion of shared SNPs linked with QTL.

**Table 9 genes-13-02100-t009:** The number of SNPs unique to one criteria or shared across two or more criteria for a genetic architecture of 1 k causative variants distributed across all chromosomes.

	Unique to One Criteria			Selected by Two Criteria			
Panel Size (k)	F${}_{\mathrm{ST}}$ Scores	*p*-Values	Fuzzy Scores	F${}_{\mathrm{ST}}$ Scores and *p*-Values	F${}_{\mathrm{ST}}$ and Fuzzy Scores	*p*-Values and Fuzzy Scores	Selected by All Criteria
1	711	205	172	80	113	619	97
2	1420	592	592	140	235	1062	206
3	2099	1119	872	147	394	1374	361
4	2662	1672	1165	151	658	1647	530
5	3137	2232	1366	146	1011	1917	706
10	4815	5075	1789	92	3378	3118	11,714
20	8937	10,110	3133	73	7050	5877	3940
30	12,957	14,736	4300	66	10,502	8723	6475
40	16,826	19,136	5333	63	13,867	11,557	9244
50	20,500	23,347	6213	60	17,194	14,348	12,245

## Data Availability

Programs used to generate the simulated data are publicly available. Programs used to analyze the data are available upon request.

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
