# Peer review of "Fuzzy Logic as a Strategy for Combining Marker Statistics to Optimize Preselection of High-Density and Sequence Genotype Data"

_genes, 2022, doi:10.3390/genes13112100_

Round 1

Reviewer 1 Report

This is a good work by the authors, they presented very well and provided enough pieces of research evidence along with tables summarizing their findings. The manuscript is well written.

Lines 23-25, Under Introduction- include citations supporting the argument.

Lines 338-354, Under conclusions, please, include suggestions for future research, highlighting the limitation of having used simulated data, and the potential drawbacks of this analysis when used with real data. 

Author Response

We thank the reviewer for the thoughtful critique regarding our manuscript.

Lines 23-25: We agree that supporting citations are necessary for this assertion and have added two citations that discuss (1) the high degree of multicollinearity and redundancy in high-density genotype data in commercial production animal populations and (2) the potential negative consequences of overparameterized genomic prediction models.

Lines 338-354: We have added to the conclusion section the limitations of simulated data as well as plans for future research utilizing real data and future model developments.

We sincerely thank the reviewer again and hope these revisions are to their satisfaction.

Reviewer 2 Report

In this manuscript, the authors present a fuzzy logic approach for single nucleotide polymorphism (SNP) marker preselection. They explain that the limited success of single-criteria SNP preselection to improve prediction accuracies is likely to be explained by the limited sensitivity of any single criteria to distinguish between relevant and non-relevant markers and, for this reason, they argue about the need of multi-criteria approaches. For this reason, the authors concatenate two prioritization criteria into a unique fuzzy score to prioritize relevant SNP markers, with the expectation of reducing the number false positives tagged by each individual criteria separately. 

In overall, this article is interesting, it is well-motivated and the methodology is clearly presented. Nevertheless, there are some aspects to be improved:

1) The state-of-art is lightly explained with mostly outdated references (very few articles published during the last three years).

2) They must to explain why they decide to use a fuzzy logic approach but no other multi-criteria strategies.

3) The validation experiments are sound, but they only use simulated data. Even when I agree that this kind of artificial data is valuable for testing several experimental scenarios, it is also important to present and discuss results with real-world data.

4) They must to present a performance comparison of their approach with other multi-criteria strategies.

Author Response

We thank the reviewer for the thoughtful critique regarding our manuscript.

1) We agree that discussion of more recent research is warranted and were able to identify 3 additional relevant articles published within the past 2 years as well as 3 articles published within the past 15 years, which have been incorporated into the introduction section at lines 61-69. Unfortunately application of fuzzy logic methodology in ours and related fields is relatively unusual and so recent relevant work is sparse.

2) We were only able to identify one other proposed strategy for multi-criteria selection of markers (Xiang et al 2019) and this strategy is not suited for simulated data. As we were proposing use of the fuzzy logic methodology in a novel context and expect that the majority of our target audience will be unfamiliar with it, we chose to focus on evaluating the fuzzy logic methodology in greater depth and rather than introduce several novel preselection strategies in one paper.

3) Unfortunately we do not currently have access to real high-density genomic data. In the conclusions section at line 362-367 we added additional discussion acknowledging the limitations of the simulated data and future research aims, including application of the proposed methodology to real data.

4) Please reference reply 2)

We sincerely thank the reviewer again and hope these revisions are to their satisfaction.

Round 2

Reviewer 2 Report

The authors satisfactory answer to all my previous concerns.